# Detection and Monitoring of Tumor-Derived Mutations in Circulating Tumor DNA Using the UltraSEEK Lung Panel on the MassARRAY System in Metastatic Non-Small Cell Lung Cancer Patients

**DOI:** 10.3390/ijms241713390

**Published:** 2023-08-29

**Authors:** Paul van der Leest, Melanie Janning, Naomi Rifaela, Maria L. Aguirre Azpurua, Jolanthe Kropidlowski, Sonja Loges, Nicolas Lozano, Alexander Sartori, Darryl Irwin, Pierre-Jean Lamy, T. Jeroen N. Hiltermann, Harry J. M. Groen, Klaus Pantel, Léon C. van Kempen, Harriet Wikman, Ed Schuuring

**Affiliations:** 1Department of Pathology (EA10), University Medical Center Groningen, University of Groningen, 9700 RB Groningen, The Netherlands; p.van.der.leest@umcg.nl (P.v.d.L.); n.rifaela@hotmail.com (N.R.); ma_aguirre1@yahoo.com.ar (M.L.A.A.); l.van.kempen@umcg.nl (L.C.v.K.); 2German Cancer Research Center (DKFZ)-Hector Cancer Institute, University Medical Center Mannheim, 68167 Mannheim, Germany; melanie.janning@medma.uni-heidelberg.de; 3Division of Personalized Medical Oncology (A420), German Cancer Research Center (DKFZ), 69120 Heidelberg, Germany; s.loges@uke.de; 4Department of Personalized Oncology, University Hospital Mannheim, Medical Faculty Mannheim, University of Heidelberg, 68167 Mannheim, Germany; 5Department of Oncology, Hematology and Bone Marrow Transplantation with Section Pneumology, University Medical Center Hamburg-Eppendorf, 20251 Hamburg, Germany; 6Department of Tumor Biology, University Medical Center Hamburg-Eppendorf, 20251 Hamburg, Germany; j.kropidlowski@uke.de (J.K.); pantel@uke.de (K.P.); h.wikman@uke.de (H.W.); 7Institut d’Analyse Génomique Imagenome, Labosud, 34070 Montpellier, France; 8Agena Bioscience GmbH, 22761 Hamburg, Germany; alexander.sartori@agenabio.com; 9Agena Bioscience, Brisbane 4006, Australia; Darryl.irwin@agenabio.com; 10Department of Clinical Research, Clinique BeauSoleil, 34070 Montpellier, France; 11Department of Pulmonary Medicine, University Medical Center Groningen, University of Groningen, 9700 RB Groningen, The Netherlands; t.j.n.hiltermann@umcg.nl (T.J.N.H.); h.j.m.groen@umcg.nl (H.J.M.G.)

**Keywords:** circulating tumor DNA, liquid biopsy, resistance mutations, progressive disease, metastatic non-small cell lung cancer

## Abstract

Analysis of circulating tumor DNA (ctDNA) is a potential minimally invasive molecular tool to guide treatment decision-making and disease monitoring. A suitable diagnostic-grade platform is required for the detection of tumor-specific mutations with high sensitivity in the circulating cell-free DNA (ccfDNA) of cancer patients. In this multicenter study, the ccfDNA of 72 patients treated for advanced-stage non-small cell lung cancer (NSCLC) was evaluated using the UltraSEEK^®^ Lung Panel on the MassARRAY^®^ System, covering 73 hotspot mutations in *EGFR*, *KRAS*, *BRAF*, *ERBB2*, and *PIK3CA* against mutation-specific droplet digital PCR (ddPCR) and routine tumor tissue NGS. Variant detection accuracy at primary diagnosis and during disease progression, and ctDNA dynamics as a marker of treatment efficacy, were analyzed. A multicenter evaluation using reference material demonstrated an overall detection rate of over 90% for variant allele frequencies (VAFs) > 0.5%, irrespective of ccfDNA input. A comparison of UltraSEEK^®^ and ddPCR analyses revealed a 90% concordance. An 80% concordance between therapeutically targetable mutations detected in tumor tissue NGS and ccfDNA UltraSEEK^®^ analysis at baseline was observed. Nine of 84 (11%) tumor tissue mutations were not covered by UltraSEEK^®^. A decrease in ctDNA levels at 4–6 weeks after treatment initiation detected with UltraSEEK^®^ correlated with prolonged median PFS (46 vs. 6 weeks; *p* < 0.05) and OS (145 vs. 30 weeks; *p* < 0.01). Using plasma-derived ccfDNA, the UltraSEEK^®^ Lung Panel with a mid-density set of the most common predictive markers for NSCLC is an alternative tool to detect mutations both at diagnosis and during disease progression and to monitor treatment response.

## 1. Introduction

Liquid biopsy approaches using biological fluids for the detection of circulating tumor cells or tumor cell-derived genomic material (i.e., DNA or RNA) offer a suitable alternative to overcome the limitations of conventional tumor tissue-based diagnostics. The analysis of circulating cell-free DNA (ccfDNA) from blood plasma has shown promising clinical value for the identification of targetable mutations and monitoring tumor response to therapy, being minimally invasive and having short turnaround times (TAT) [1,2]. Current clinical applications of ccfDNA-based analysis encompass treatment guidance in non-small cell lung cancer (NSCLC) through identification of, e.g., *EGFR*, *BRAF*, and *KRAS* mutations both at primary diagnosis [3] and progression upon treatment resistance [4]. Previous studies showed that monitoring serial ccfDNA samples during treatment has predictive value for both early tumor response and durable clinical benefit, which is important for accurate treatment decision-making [5,6,7,8].

Currently, many diagnostic laboratories use molecular tumor tissue profiling in their routine to detect therapeutically targetable mutations. Accurate detection of molecular predictive profiles in pretreatment tumor tissue biopsies and acquired resistance mutations in recurrent disease is crucial for suitable patient care. Despite the advances in reliable tumor tissue-based molecular profiling, the invasive nature of collecting tumor biopsies hinders repetitive sample collection and, thereby, optimal disease monitoring and early identification of resistance mechanisms. In addition, in daily practice, molecular predictive testing in tumor tissue is regularly not feasible due to insufficient or inadequate material, making liquid biopsy testing relevant [9]. Liquid biopsy approaches show great promise for accurate variant calling in circulating tumor DNA (ctDNA) with sufficient analytical potential to surmount the shortcomings of tumor tissue-based testing [10]. Early detection of treatment resistance mutations enables therapy adjustments to halt progressive disease and extend survival. Particularly in *EGFR*-mutated NSCLC, treatment with first-line tyrosine kinase inhibitors (TKI) frequently results in the development of primary and secondary resistance mutations (e.g., *EGFR* p.(T790M) and *EGFR* p.(C797S), respectively) that might act as targets for other treatment options [11,12].

Accumulating reports support the applicability of the ctDNA fraction of ccfDNA for molecular tumor profiling both at primary diagnosis and at progression [1,9,13]. In general, recurrent challenges of tissue biopsy, such as intra- and inter-tumor heterogeneity and tumor accessibility, can be resolved using ccfDNA-based molecular tumor profiling [9,14]. However, plasma ctDNA represents a low percentage (<1%) of the total ccfDNA [1]. Due to recent advances in blood-based mutation-specific analysis techniques based on polymerase chain reaction (PCR) and next-generation sequencing (NGS), tumor-derived genetic aberrations are detectable with increasing sensitivity and specificity when optimal preanalytical conditions of plasma and ccfDNA processing are guaranteed [15,16,17]. Droplet digital PCR (ddPCR) is one of the most sensitive and least expensive methods for the analysis of low copies of mutant ctDNA [18]. However, each ddPCR assay presently allows the detection of a single or few mutations. Therefore, ddPCR is not the preferred method for routine testing of numerous mutations but holds promise for monitoring and predicting treatment response with tumor-informed selected variants [7,19]. For predictive testing in tumor tissue, broader NGS panels that cover the most common predictive markers are used in current routine clinical practice [20]. Since guidelines recommend elaborate molecular tumor profiling to identify actionable mutations, ccfDNA-based versions of these NGS panels with sufficient analytical sensitivity have significant clinical value [21,22,23]. However, on-site NGS of large gene panels on plasma-derived ccfDNA (e.g., Avenio ctDNA Expanded Kit) or commercial centralized testing (i.e., FoundationOne Liquid CDx, Guardant360 CDx) is cost-prohibitive and not reimbursed in most countries, with a long TAT (a minimum of 5–10 days) and the need for expertise to perform complex sequence analysis and proper variant calling [14,23,24]. Analysis of smaller NGS panels (i.e., Oncomine Lung cfDNA Assay and QIAseq Human Actionable Solid Tumor Panel) reduces the TAT and facilitates clinical interpretation; however, they remain cost-prohibitive as long as sequencing costs are not reduced. Targeted panels covering the majority of current actionable mutations therefore have clinical value as cost-effective alternatives with short processing times [23,25]. PCR-based assays, generally encompassing small panels targeting hotspot mutations in a single gene, have been developed as well. To date, only two PCR-based liquid biopsy tests have FDA approval as companion diagnostic assays for selecting TKI treatment in *EGFR*-mutated NSCLC (i.e., Cobas^®^ EGFR Mutation Test v2 and Therascreen^®^ EGFR RGQ PCR) [26]. Targeted treatment options are not limited to mutated *EGFR*; therefore, broader plasma mutation screening panels are required.

The UltraSEEK^®^ Lung Panel on the MassARRAY^®^ System is an assay specifically designed for the detection of multiple clinically relevant NSCLC-associated mutations in plasma. A recent comparison study showed comparable sensitivity and specificity in *EGFR* mutation detection between the PCR-based Cobas^®^ EGFR Mutation Test v2 and UltraSEEK^®^ Lung Panel, especially with an optimal ccfDNA input for UltraSEEK^®^ (≥10 ng) [27]. Furthermore, it was demonstrated previously that clinically relevant variants beyond *EGFR* could be identified as well [27,28], including in patients with limited oligo–brain metastatic disease [29]. Within the framework of the Innovative Medicines Initiative (IMI) program CANCER-ID (http://www.cancer-id.eu, accessed on 14 June 2023), the UltraSEEK^®^ Lung Panel on the MassARRAY^®^ System to detect tumor-derived driver mutations in ccfDNA of NSCLC patients at diagnosis and at progression was evaluated, as well as its ability to quantify changes in ctDNA levels during treatment as a monitoring tool.

## 2. Results

### 2.1. The Limit of Detection of the UltraSEEK^®^ Lung Panel Using Reference Material

An international multicenter evaluation by three laboratories participating in the CANCER-ID consortium evaluated the limit of detection of the UltraSEEK^®^ Lung Panel using Seraseq^®^ ctDNA Complete™ Mutation Mix reference material. Reference samples contain ten clinically relevant mutations at different VAFs (0.1, 0.5, 1.0, and 2.5%) covered by the UltraSEEK^®^ Lung Panel. Different DNA input amounts (5, 10, and 20 ng) were analyzed. At a VAF of ≥1%, an overall detection rate of over 99% was observed for all DNA input amounts (Figure 1A; Appendix A). The detection rate drops to >90% for 0.5% VAF and >50% for 0.1%, irrespective of ctDNA input. VAFs measured by UltraSEEK^®^ were comparable with the expected VAFs for all DNA inputs (Figure 1B). Twenty-two false-positive calls were identified in a total of 5022 variant positions, resulting in a variant level specificity of >99.5% (Appendix A). False-positive variants were observed across eleven samples (88% sample-level specificity), of which seven had a DNA input of 5 ng.

### 2.2. Comparison of Detection of Mutant ctDNA in Patient-Derived Plasma between UltraSEEK^®^ and ddPCR

In total, 131 variants were detected across 45 patients at any timepoint using UltraSEEK^®^. Sixty-one (47%) were identified in *EGFR*, 38 (29%) in *KRAS*, 21 (16%) in *BRAF*, 10 (8%) in *PIK3CA*, and 1 (1%) in *ERBB2*. An independent mutant ctDNA detection method (i.e., mutation-specific ddPCR analysis) was performed on the same plasma for 157 samples to determine the agreement between both assays. Twelve discordant samples were observed that were either negative with UltraSEEK^®^ (*n* = 5) or ddPCR (*n* = 7), resulting in an overall concordance of 92% (145/157) with a PPA of 93% and an NPA of 91% (Appendix A).

### 2.3. Detection of Tumor Tissue-Derived Variants in the Baseline Plasma Sample Using the UltraSEEK^®^ Lung Panel

Data on mutational profiles from routine diagnostic NGS analysis on the pretreatment tissue biopsy was available for 66 matched tumor tissue samples (92%). Of the 77 diagnostically or clinically relevant variants in *BRAF*, *EGFR*, *ERBB2*, *KRAS*, and *PIK3CA* reported in these pretreatment tissue samples, 53 variants were detected in plasma at baseline using UltraSEEK^®^ (Appendix A). Eighteen (23%) tumor tissue mutations were not detected in the plasma, and three (4%) of the mutations detected in the plasma were not present in the tumor specimen. CcfDNA UltraSEEK^®^ analysis showed concordant results with tumor tissue NGS analysis in 68% of the patients for the mutations covered by the UltraSEEK^®^ Lung Panel (Figure 2A). Nine variants reported in the tumor tissue across *EGFR*, *KRAS*, *BRAF*, and *PIK3CA* were not covered by the UltraSEEK^®^ Lung Panel (Figure 2B). Restricting the comparison to therapeutically targetable mutations only covered by the UltraSEEK^®^ Lung Panel, an 80% concordance between tumor tissue NGS and UltraSEEK^®^ was observed (Figure 2C). A considerably lower concordance was found for non-actionable mutations in NSCLC (52%). One mutation (*KRAS* c.34_35delinsTT; p.(G12F)) called in the tumor affected the same genomic position in plasma but with a different annotation (*KRAS* c.34G>T; p.(G12C)), most probably due to the inability of the platform to differentiate between covered variants and certain complex variants at similar coding DNA sequences (Appendix A).

In eleven patients, none of the variants previously reported in tissue samples were identified with UltraSEEK^®^ ccfDNA analyses in the baseline plasma sample. To determine whether these were truly non-shedding tumors, the same plasma was reevaluated with ddPCR as an independent analytically sensitive mutant ctDNA detection assay (Appendix A). Twelve of fourteen variants in 9/11 patients were not detected with ddPCR as well, whereas two cases harboring *KRAS* mutations were detected (Appendix A). Two out of 69 variants reported in the tumor tissue and detected with UltraSEEK^®^ were incorrectly missed in the baseline plasma sample, leading to a 3% false-negative rate at baseline.

### 2.4. Survival Analysis According to ctDNA Dynamics Using UltraSEEK^®^

When comparing the mutant molecule levels at start of treatment (t_0_) and first response evaluation (t_1_), patients were categorized as ctDNA decrease (a ≥15% lower VAF at t_1_, *n* = 18), ctDNA increase (a ≥ 15% higher VAF at t_1_, *n* = 12), ctDNA negative (no detectable ctDNA at both timepoints, *n* = 27), or ctDNA stable (change in VAF < 15% at t_1_, *n* = 5); decreased ctDNA levels were associated with improved clinical outcome (Figure 3A,B). Comparing all patients with increasing or stable ctDNA levels (*n* = 17) against those with decreased ctDNA levels (*n* = 21), a decrease was associated significantly with longer median survival and a reduced hazard ratio (HR): PFS (46 versus 6 weeks; HR: 0.47; *p* < 0.05) and OS (145 versus 30 weeks; HR: 0.36; *p* < 0.01) (Figure 3C,D).

For 41 patients, an observed change in VAF (UltraSEEK^®^) was compared with the change in mutant molecule levels (ddPCR) between t_0_ and t_1_. Stepwise Cox regression analysis resulted in significantly prolonged PFS and OS upon decreased ctDNA levels, but with lower hazard ratios for PFS with ddPCR (0.32) than UltraSEEK^®^ (0.48) (Appendix A). Concordant dynamics were observed in 85% of the cases (Appendix A; Appendix A). In six discordant cases, the number of mutant molecules measured with ddPCR was very low in all cases (less than 55 copies/mL of plasma, equivalent to <10 mutant droplets), of which either the UltraSEEK^®^ or ddPCR was negative at both timepoints in plasma in four cases (Appendix A).

### 2.5. Plasma-Based Determination of Disease Progression and Acquired Treatment Resistance

In our cohort of 72 patients, twelve of thirteen were progressive on TKI and 48 of 59 on other treatment modalities. Plasma samples at progression were available for 40 patients. Mutations detected in ccfDNA at progression only and associated with treatment resistance were identified in four patients with an activating *EGFR* mutation in the pretreatment tumor biopsy and treated with first-line EGFR-TKI (Appendix A). Two patients developed the *EGFR* p.(T790M) mutation following afatinib treatment, and two patients acquired the *EGFR* p.(C797S) mutation during osimertinib therapy. In all four patients, these mutations were not detected in any of the earlier plasma samples.

## 3. Discussion

In the absence of tumor tissue biopsy specimens for predictive testing or during treatment response monitoring, there is a need to accurately detect tumor-derived mutations in a blood sample with high sensitivity. Advancements in ultrasensitive ctDNA detection offer new opportunities for applications of mutation detection in cell-free plasma for personalized treatment management [1]. Targeted panel analyses such as UltraSEEK^®^ are a cost-efficient alternative to detecting most of the current actionable mutations. Previous comparative analysis revealed that targetable mutation screening for an equal number of NSCLC patients reduced the total costs by approximately two-thirds compared to NGS [25]. Therefore, the clinical utility of the UltraSEEK^®^ test for the detection of mutations in plasma from patients with metastatic NSCLC was assessed in this multicenter evaluation.

UltraSEEK^®^ analysis of reference material (Seraseq^®^ ctDNA Complete™ Mutation Mix) demonstrated an overall high detection rate for ten different mutations, irrespective of ctDNA input, with 22 false-positive calls, a variant level specificity of >99.5%, and a sample level specificity of 88%. False-positive results, however, were predominantly observed in low-input samples (5 ng), supporting the use of the recommended ≥10 ng input for accurate results. The analysis of 125 plasma samples from 37 NSCLC patients revealed an overall concordance of 90% with a PPA of 91% and an NPA of 88% between UltraSEEK^®^ and ddPCR. Similar studies with several types of cancer identified an 88% and 80% concordance between these two platforms [30,31].

The availability of tumor tissue remains today’s best clinical practice and is essential for the initial molecular characterization of cancer. However, when tissue is not available or adequate for NGS analysis, tumor-derived mutation testing in plasma serves as an alternative method. Our results demonstrated concordant results in 68% of the patients between routine diagnostic NGS analysis on the pretreatment tissue biopsy and UltraSEEK^®^ analysis in 66 matched baseline plasma samples, comparable to previous data [28,32]. In 23% of the patients with a known driver mutation in the tumor tissue, this variant could not be detected in the matched baseline plasma sample. This discrepancy is in line with previous reports [13,28]. Increased clearance, a short half-life, or a lack of shedding of tumor-specific DNA into the circulation were suggested causes for the absence of ctDNA in pretreatment plasma samples [1]. Reevaluation of negative plasma samples with highly sensitive ddPCR analysis revealed that nine of eleven ctDNA-negative patients were true negatives. A lack of sufficient analytical sensitivity of UltraSEEK^®^ and ddPCR to detect ctDNA levels below 0.5% VAF cannot be excluded.

Elaborate molecular tumor profiling using an NGS panel could elevate the number of ctDNA-positive patients by analyzing more targets simultaneously [6]. However, it does not necessarily identify more actionable targets for patients with advanced-stage NSCLC. Recently, Aggarwal et al. reported on a ccfDNA-based NGS assay with over 74 cancer-related genes and identified 113 variants in 323 patients with metastatic NSCLC. Only 42 patients were treated with targeted therapy as a result of the detection of mutations in *BRCA1* (*n* = 1), *ALK* (*n* = 1 fusion), *MET* exon14 skipping (*n* = 4), *BRAF* p.(V600E) (*n* = 2), and *EGFR* (*n* = 34) (20). Theoretically, mutations in 36 of these 42 patients (85%) would have been detected with UltraSEEK^®^ as well. When restricting the data of the present study to therapeutically targetable mutations covered by the panel only, the concordance with tumor tissue NGS at baseline was 80%. Recent studies comparing NGS of tumor tissue biopsies and matched plasma at baseline reported a comparable overall concordance of 81% [6,21] and up to 91% for actionable variants only [33]. In our cohort, nine diagnostically relevant variants reported in tumor tissue were not covered by the platform. These data imply that, in cases of inadequate tumor tissue, UltraSEEK^®^ enables sensitive genotyping and covers >80% of the common predictive markers detected in tumor tissue of advanced NSCLC used in routine diagnostic settings. Furthermore, complementary ccfDNA molecular profiling might display the actual mutational spectrum of patients, which may provide insights on predictive mutations missed during molecular tumor profiling, identify acquired low copy resistance mechanisms (i.e., secondary mutations), or show inter- or intratumor heterogeneity in patients with discordant clinical responses [23]. When the analysis renders the result negative, however, subsequent testing for actionable mutations and fusions is required to cover all demanded targets in accordance with guidelines. Prospective studies to prove the clinical utility of complementary ccfDNA molecular testing using targeted panels are urgently needed.

Cancer progression as a result of resistance to targeted therapy is often accompanied by the detection of novel mutations or clonal expansion of non-responsive tumor cells [34,35]. For instance, on- and off-target resistance mutations after treatment with the EGFR inhibitor osimertinib include de novo *EGFR* p.(C797S), *BRAF* p.(V600E), *PIK3CA* p.(H1047), and *KRAS* mutations [35,36,37]. Despite advances in treatment guidance, ctDNA-based companion testing for resistance mutations remains limited to *EGFR* testing in routine diagnostics.

The analytical sensitivity to detect *EGFR* mutations in the cell-free plasma of NSCLC patients with methods such as Cobas^®^ [38], ddPCR [9], and ARMS [39] revealed a pooled sensitivity of 67% and a specificity of 94% [40]. A high concordance of 86% between UltraSEEK^®^ and Cobas^®^ for the detection of *EGFR* mutations in a cohort of 137 patients who progressed under EGFR-TKI treatment was reported recently [27]. Importantly, resistance mechanisms that cannot be detected with the Cobas^®^ test, including *KRAS* mutations and the *EGFR* p.(C797S) mutation, were detected in eighteen cases [27]. In the present study, in four out of twelve patients who progressed under EGFR-TKI, on-target resistance mutations were detected with UltraSEEK^®^. UltraSEEK^®^ enables sensitive genotyping for the detection of the most common EGFR-TKI resistance mutations in plasma with the advantages of ease of use, low costs, and a short TAT. As such, ccfDNA molecular profiling using UltraSEEK^®^ is an alternative to the FDA-approved Cobas^®^ EGFR Mutation Test v2 assay. However, the current panel is not useful for the detection of resistance toward and immune checkpoint inhibitor therapy (representing 75% of the entire cohort) since mutations in *KEAP1* and *STK11* are not covered [41], as well as TKI other than EGFR-TKI (e.g., *ALK*) [42].

Longitudinal analysis of ccfDNA has shown clinical utility in the identification of early progression and durable responses using single [7,43] and multiple target approaches [6,8,44]. Here, a decrease in ctDNA over time measured with UltraSEEK^®^ was similarly associated with prolonged PFS and OS. However, quantitative ddPCR analysis, regarded as a robust early response assessment tool [7,45], better predicts PFS, resulting in a lower HR. Nonetheless, semiquantitative UltraSEEK^®^ analysis successfully identified patients with durable treatment responses, illustrating its potential as a monitoring tool (Appendix A).

In conclusion, ideal blood-based tests to primarily diagnose cancer patients and monitor the clonal evolution of their tumors would identify tumor-derived biomarkers with ultra-sensitivity, be fast, and be cost-efficient. Cost-efficiency is mainly determined by testing a limited number of clinically relevant variants during the follow-up of patients, optimized for a specific cancer type. Such an approach covers the known on-target and off-target resistance variants and must be flexible to incorporate future novel variants. The MassARRAY^®^-based UltraSEEK^®^ Lung Panel enables rapid, sensitive, and semiquantitative genotyping for 73 relevant mutations in *BRAF*, *EGFR*, *ERBB2*, *KRAS*, and *PIK3CA*. Although a tumor tissue biopsy remains essential for initial cancer diagnosis, the UltraSEEK^®^ Lung Panel serves as a molecular profiling tool to detect the most common actionable targets in the absence of adequate tumor tissue DNA. In the event that UltraSEEK^®^ results turn out negative, subsequent molecular profiling—including genomic rearrangements, e.g., *ALK* and *MET* exon14 skipping—is necessary to cover all actionable targets. Future prospective studies are required to explore the clinical utility of UltraSEEK^®^ versus comparable ctDNA detection assays at diagnosis and disease progression.

## 4. Materials and Methods

### 4.1. Sample Collection

For patient inclusion, a cell-free plasma sample should at least have been available at the start of treatment (t_0_), preferably at the first response evaluation (t_1_, 4–6 weeks after the start of treatment), and, when applicable, at the presentation of disease progression (t_p_). In addition, the inclusion of patients with NSCLC who were tested for the presence of predictive mutations in the pretreatment tissue biopsy was preferred. Patients were treated for NSCLC at either the University Medical Center Groningen (UMCG cohort) or the University Medical Center Hamburg-Eppendorf (UKE cohort). In total, 177 cell-free plasma samples from 72 NSCLC patients were included, of which 72 were at t_0_, 65 at t_1_, and 40 at t_p_. The clinical and pathological characteristics are summarized in Table 1. The majority of patients (*n* = 54) were treated with immunotherapy as no targetable mutations were detected in the pretreatment tumor biopsy, while most of the *EGFR*-mutated tumors were treated with TKIs (*n* = 13). Other patients received chemotherapy either alone or in combination with immunotherapy. Plasma sample processing was performed as described in the Appendix A.

### 4.2. Molecular Analysis

CcfDNA was extracted from blood plasma as described previously [7,17,46] (see Appendix A). Tumor tissue NGS results were retrieved from the Dutch nationwide pathology registry (PALGA) for relevant predictive variants in *BRAF*, *EGFR*, *ERBB2*, *KRAS*, and *PIK3CA*. The UltraSEEK^®^ Lung Panel covers 73 hotspot mutations in these five genes on the matrix-assisted laser desorption ionization time-of-flight (MALDI-TOF)-based MassARRAY^®^ System (Appendix A). Molecular profiling was performed at t_0_, t_1_, and t_p_ using the UltraSEEK^®^ Lung Panel on the MassARRAY^®^ System [24,29,31,47,48] (Agena Bioscience, San Diego, CA, USA) and ddPCR [7,17] (Bio-Rad Laboratories, Pleasanton, CA, USA) according to the manufacturer’s recommendations (for details, see Appendix A; Appendix A). With respect to the dynamics of ctDNA levels, changes in variant allele frequency (VAF) equal to or greater than 15% (UltraSEEK^®^) and changes in mutant molecule levels equal to or greater than 31% (ddPCR) between t_0_ and t_1_ were considered an increase or decrease (for details, see Appendix A).

### 4.3. Statistical Analysis

Descriptive statistics were used for patient and tumor characteristics. Clopper–Pearson binomial confidence intervals were calculated for the limit of detection. For statistical assessment between different ccfDNA inputs, the Kruskal–Wallis test was performed, followed by Dunn’s multiple comparison test. Agreement between UltraSEEK^®^ analysis and either tumor tissue NGS or ddPCR analysis on ccfDNA was expressed as concordance, positive percent agreement (PPA), or negative percent agreement (NPA). Progression-free survival (PFS) and overall survival (OS) were defined as the period between the date of the start of therapy and the date of progressive disease or death, respectively. The data were censored at the date of the last follow-up in the absence of an event. Kaplan–Meier survival data were stratified for mutant ctDNA data and compared with the log-rank test. Radiological reports and liquid biopsy test results were assessed independently. GraphPad Prism 8.4.2 software was used for all statistical analyses. A *p*-value < 0.05 was considered significant.

## Figures and Tables

**Figure 1 ijms-24-13390-f001:**
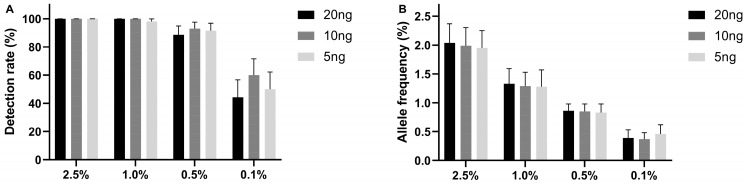
Detection of hotspot mutations common in NSCLC using UltraSEEK^®^ Lung Panel. (**A**) Detection rate and (**B**) quantification of Seraseq^®^ ctDNA Complete™ Mutation Mix reference material at variant allele frequencies (VAFs) of 2.5%, 1.0%, 0.5%, and 0.1% were analyzed using 20 ng (black), 10 ng (dark grey), and 5 ng (light grey) ccfDNA input. Error bars represent the confidence intervals calculated from the detection rates per mutation; detailed results of ten hotspot mutations in *EGFR*, *KRAS*, *BRAF*, *ERBB2*, and *PIK3CA* are presented in Appendix A. Kruskal–Wallis test was performed, followed by Dunn’s multiple comparison test; all comparisons were not significant.

**Figure 2 ijms-24-13390-f002:**
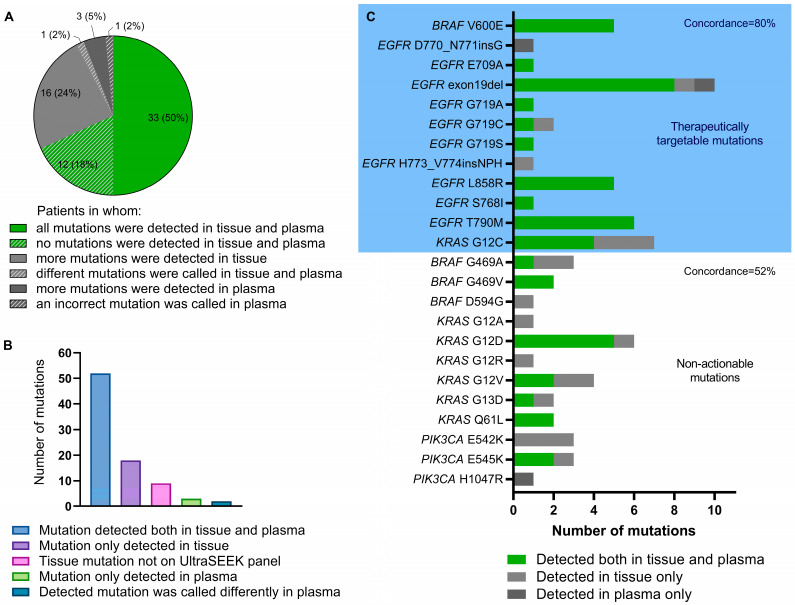
Concordance between mutations detected in tumor tissue and plasma-derived ccfDNA in *EGFR*, *KRAS*, *BRAF*, *ERBB2*, and *PIK3CA*. (**A**) Pie chart representing the patients in whom mutation detection using tissue NGS and plasma UltraSEEK^®^ Lung Panel analyses was considered concordant (green) and discordant (grey) for all mutations covered by the UltraSEEK^®^ Lung Panel. (**B**) Bar chart representing the distribution of all mutations detected using tumor tissue NGS reported in the pathology archives and plasma UltraSEEK^®^ Lung Panel analysis. (**C**) Stacked bar chart displaying the distribution of individual mutations detected both with tissue NGS and plasma UltraSEEK^®^ analysis (green), only with tissue NGS (light grey), and only with plasma UltraSEEK^®^ (dark grey). Clinically relevant mutations (light blue box) were separated from currently non-actionable mutations in non-small cell lung cancer (white box). Individual patient results and mutations not covered by the UltraSEEK^®^ Lung Panel are depicted in Appendix A.

**Figure 3 ijms-24-13390-f003:**
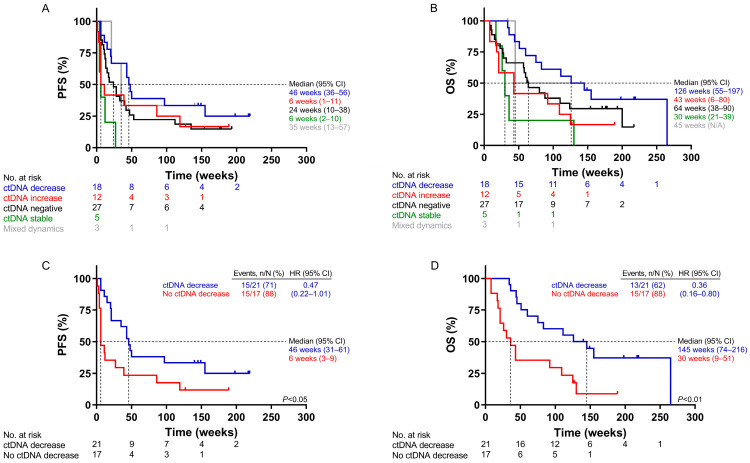
Survival analysis according to ctDNA dynamics using the UltraSEEK^®^ Lung Panel. Kaplan–Meier plot displaying the (**A**) progression-free survival (PFS) and (**B**) overall survival (OS) of patients with decreasing (blue), negative (black), stable (green), or increasing (red) ctDNA levels. (**C**) PFS and (**D**) OS of patients with decreasing ctDNA levels (blue) or no decrease in ctDNA (red). No decrease in ctDNA encompasses all patients without a >15% decrease in ctDNA. Horizontal dashed lines indicate the median survival; vertical lines indicate the moment at which the median survival was reached. HR, hazard ratio; CI, confidence interval. Log-rank test, *p*-value < 0.05, was considered significant.

**Table 1 ijms-24-13390-t001:** Clinical and pathological characteristics.

	Total Cohort	UMCG Cohort	UKE Cohort
**Patients**	72	60	12
**Median age**	65 (38–85)	63 (38–85)	68 (57–79)
**Sex**			
Male	37 (51%)	29 (48%)	8 (67%)
Female	35 (49%)	31 (52%)	4 (33%)
**ECOG PS**			
0	23 (32%)	21 (35%)	2 (17%)
1	44 (61%)	37 (62%)	7 (58%)
2	5 (7%)	2 (3%)	3 (25%)
**Histology**			
Adenocarcinoma	64 (89%)	56 (93%)	8 (67%)
Squamous cell carcinoma	8 (11%)	4 (7%)	4 (33%)
**Smoking status**			
Active smoker	33 (46%)	29 (48%)	4 (33%)
Former smoker	27 (38%)	23 (38%)	4 (33%)
Never smoker	11 (15%)	8 (13%)	3 (25%)
Unknown	1 (1%)		1 (8%)
**Current treatment**			
*Chemotherapy*	*3 (4%)*	*1 (2%)*	*2 (17%)*
Carboplatine/pemetrexed	1 (1%)		1 (8%)
Cisplatine/pemetrexed	1 (1%)	1 (2%)	
Docetaxel/ramucirumab	1 (1%)		1 (8%)
*Chemo-immunotherapy*	*2 (3%)*	*1 (2%)*	*1 (8%)*
Carboplatine/paclitaxel/bevacizumab	1 (1%)	1 (2%)	
Carboplatine/pemetrexed/pembrolizumab	1 (1%)		1 (8%)
*Immunotherapy*	*54 (75%)*	*45 (75%)*	*9 (75%)*
Atezolizumab	1 (1%)	1 (2%)	
Nivolumab	40 (56%)	37 (62%)	3 (25%)
Pembrolizumab	13 (18%)	7 (12%)	6 (50%)
*Targeted therapy*	*13 (18%)*	*13 (22%)*	
Afatinib	7 (10%)	7 (12%)	
Gefitinib	1 (1%)	1 (2%)	
Osimertinib	5 (7%)	5 (8%)	
**Previous lines of therapies**			
0	19 (26%)	11 (18%)	8 (67%)
1	32 (44%)	30 (50%)	2 (17%)
2	15 (21%)	15 (25%)	
≥3	6 (8%)	4 (7%)	2 (17%)

ECOG PS, Eastern Cooperative Oncology Group performance-status score.

## Data Availability

Data is available upon reasonable request.

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
