# Peer review of "Detection and Monitoring of Tumor-Derived Mutations in Circulating Tumor DNA Using the UltraSEEK Lung Panel on the MassARRAY System in Metastatic Non-Small Cell Lung Cancer Patients"

_ijms, 2023, doi:10.3390/ijms241713390_

Round 1
Reviewer 1 Report
Van der Leest, Janning and colleagues report here the results of a study assessing the accuracy of hotspot mutation calls from circulating tumor DNA of NSCLC patients, using a commercial platform based on mass spectrometry. The results were highly concordant with those obtained with ddPCR and NGS. A ctDNA level decrease was implicated with prolonged patient survival. The commercial ties of some of the authors to the company producing the platform are correctly reported in the Conflicts of Interest and Funding statements.
The manuscript is clear, well written and the data support the conclusions drawn by the authors. The quality assessment of the performance of this commercial platform is useful and important for potential users.
The scientific novelty brought to the field by this study could be improved with some additional analyses. The quantification of ctDNA increase/decrease upon treatment is intriguing as it can provide an additional quantitative way to assess treatment efficacy. Although the number of cases is low, it would be interesting if the following two points could be tested:
1) Is there any overall difference in ctDNA increase/decrease for different treatments (e.g. chemotherapy vs immunotherapy vs targeted therapy), ideally accounting for clinical assessment of treatment efficacy?
2) How does ctDNA increase/decrease compare with clinical assessment of the treatment efficacy? i.e. are decreasing ctDNA levels significantly associated with a positive clinical treatment response (assessing with e.g. Fisher test)? is this association different when comparing different treatments (e.g. immunotherapy vs targeted therapy)
Additionally, the following minor points/amendments should be addresses as well (I indicate the line number/figure/table):
35 ddPCR not defined: while readers are familiar with the acronym PCR, the meaning of the prefix 'dd' should be made explicit here
Fig 1A - why 10 ng, a smaller amount, seems to work better than 20 ng at lower VAFs? Could you provide any explanation of this?
Fig 1B - why the 2.5% VAF is underestimated and 1-0.5-0.1% VAFs are overestimated in your results? Could you provide any explanation of this?
Table 1: in the 'Current Treatment' section, please check the numbers related to 'Immunotherapy' and 'Targeted therapy', as they seem to be misaligned
Fig 2C: ERBB2 mutations are absent in this barplot, where they never found in this cohort?
197 t0/t1 not defined: although t0 and t1 are correctly defined in the Materials and Methods (at the end of the article), it would help the reader if they were briefly defined also here before being referenced
205 why the number of ctDNA-decreased patients here is 21 and not 18 (line 201)?
215 remove the comma ("No decrease in ctDNA,")
Fig 3A-3B - the presence of circulating tumor DNA is usually associated with more aggressive lung tumors (see e.g. Assaf et al. Nature Medicine 2023), while in this cohort the survival of ctDNA-negative patients seems comparable to those with ctDNA. Could the authors provide any reason for this?
265 should be "half-life"
Author Response
Van der Leest, Janning and colleagues report here the results of a study assessing the accuracy of hotspot mutation calls from circulating tumor DNA of NSCLC patients, using a commercial platform based on mass spectrometry. The results were highly concordant with those obtained with ddPCR and NGS. A ctDNA level decrease was implicated with prolonged patient survival. The commercial ties of some of the authors to the company producing the platform are correctly reported in the Conflicts of Interest and Funding statements.
The manuscript is clear, well written and the data support the conclusions drawn by the authors. The quality assessment of the performance of this commercial platform is useful and important for potential users.
The scientific novelty brought to the field by this study could be improved with some additional analyses. The quantification of ctDNA increase/decrease upon treatment is intriguing as it can provide an additional quantitative way to assess treatment efficacy. Although the number of cases is low, it would be interesting if the following two points could be tested:
1) Is there any overall difference in ctDNA increase/decrease for different treatments (e.g. chemotherapy vs immunotherapy vs targeted therapy), ideally accounting for clinical assessment of treatment efficacy?
Answer to reviewer: We thank the reviewer for this comment as this question is very relevant. We have attempted to perform this analysis, however this yielded no statistically significant results. This is probably due to the low number of cases per treatment group, high diversity in treatment lines, and the different treatment efficacies per treatment modality.
We would like to stress that the purpose of this study was to evaluate the UltraSEEK Lung Panel in an unselected patient cohort with metastasized NSCLC, hence the mixed patient population with or without (targetable) mutations and consequently different treatment strategies.
2) How does ctDNA increase/decrease compare with clinical assessment of the treatment efficacy? i.e. are decreasing ctDNA levels significantly associated with a positive clinical treatment response (assessing with e.g. Fisher test)? is this association different when comparing different treatments (e.g. immunotherapy vs targeted therapy)
Answer to reviewer: Again, a very relevant question. We have performed a Fisher’s exact test on the relation between ctDNA increase/decrease and (non)-durable response to treatment as defined by a PFS >26 weeks (now added to the legend of Supplemental Table 7), which was not significant. Also for this comparative analysis between treatment regimens the groups were too small. When only including patients treated with immunotherapy the results remained not significant.
Additionally, the following minor points/amendments should be addresses as well (I indicate the line number/figure/table):
35 ddPCR not defined: while readers are familiar with the acronym PCR, the meaning of the prefix 'dd' should be made explicit here
Answer to reviewer: We thank the reviewer for this notion and defined this abbreviation in the abstract as well.
Fig 1A - why 10 ng, a smaller amount, seems to work better than 20 ng at lower VAFs? Could you provide any explanation of this?
Answer to reviewer: We have assessed the differences between the DNA inputs (5 10 and 20ng) for all VAFs using Kruskal-Wallis test was performed followed by Dunn’s multiple comparison test (now added to the figure legend) and none of these were statistically significant. Visually, for 0.5% and 0.1%, 10ng seems to perform slightly better than 20ng. However, for the 0.5% VAF samples, not recovered variants at 20ng but detected with 10ng input were all from the same run, implicating lower quality of that specific run.
As is indicated in Supplemental Table 1, the sensitivity of each assay is 0.125% at the lowest, therefore identification of the variants at 0.1% VAF is only by chance. Input has no effect on increase this chance. Since the differences were not significant, we only mention that the detection rate drops to lower levels at 0.5 and 0.1% VAF irrespective of DNA input (lines 147-149).
Fig 1B - why the 2.5% VAF is underestimated and 1-0.5-0.1% VAFs are overestimated in your results? Could you provide any explanation of this?
Answer to reviewer: As mentioned in the Supplemental Materials and Methods, the VAFs determined by UltraSEEK are based on normalizing the mutant peak intensity against the linear regression curve of the capture controls. These controls represent a 1% VAF. In addition, assays specific correction coefficients are applied that that are derived from titration experiments down from the LOD to 2% VAF. Taken together, it can be expected that particularly above 2% VAF the effect of extrapolation is skewing the calculated VAF slightly toward the observed phenomenon. However, in this study we assessed whether the relative differences in VAF for a patient harbored clinical value. Therefore, to assess the accuracy of the titration curves was beyond the scope of this study.
Table 1: in the 'Current Treatment' section, please check the numbers related to 'Immunotherapy' and 'Targeted therapy', as they seem to be misaligned
Answer to reviewer: We have noticed the misalignment ourselves as well. However, this is only apparent in the pdf for review and not in the original manuscript. We are confident that the misalignment will be resolved during editorial formatting.
Fig 2C: ERBB2 mutations are absent in this barplot, where they never found in this cohort?
Answer to reviewer: In Figure 2, all variants identified in tumor tissue or baseline plasma samples are presented. Indeed, no ERBB2 mutations were detected in tumor tissue or plasma at baseline. In one patient, an ERBB2 mutation was identified during treatment in the t1 plasma sample (UKE-AB-011; see Supplemental Table 7).
197 t0/t1 not defined: although t0 and t1 are correctly defined in the Materials and Methods (at the end of the article), it would help the reader if they were briefly defined also here before being referenced
Answer to reviewer: We have adjusted the text accordingly (see lines 205-206).
205 why the number of ctDNA-decreased patients here is 21 and not 18 (line 201)?
Answer to reviewer: In Figure 3A/B, 3 patients with mixed ctDNA dynamics are presented. In the Supplemental Materials and Methods, the following is mentioned regarding cases with mixes ctDNA dynamics: “In cases with two or more variants and contradictory ctDNA dynamics (referred to as mixed dynamics; n=3), the variant or sum of variants with the highest VAF was considered in the grouped analysis”. Supplemental Table 7 shows that all cases with mixed dynamics (indicated with ‡) all are considered as ctDNA decreases. Therefore, the 21 patients with ctDNA decreases are included in the grouped analysis in Figure 3 C/D.
215 remove the comma ("No decrease in ctDNA,")
Answer to reviewer: We have adjusted the text accordingly.
Fig 3A-3B - the presence of circulating tumor DNA is usually associated with more aggressive lung tumors (see e.g. Assaf et al. Nature Medicine 2023), while in this cohort the survival of ctDNA-negative patients seems comparable to those with ctDNA. Could the authors provide any reason for this?
Answer to reviewer: UltraSEEK is a targeted panel covering only 73 specific mutations in five clinically relevant genes. As is mentioned several times in the discussion section, targeted panels with limited coverage of the genome have a lower chance of identifying ctDNA molecules compared to elaborate sequencing panels (e.g., FoundationONE Liquid CDx applied in Assaf et al. Nat. Med. 2023). Therefore, when a mutation is not called during targeted panel analysis (in this case UltraSEEK), it could not be determined whether this is due to not shedding ctDNA by the tumor or the inability to recover that specific variant by the assay. Therefore, we regarded ctDNA negative patients as a separate group and excluded them from the grouped survival analysis in accordance with our previous work (Leest et al. Mol. Oncol. 2021).
265 should be "half-life"
Answer to reviewer: We have adjusted the text accordingly.
Reviewer 2 Report
In this interesting study, authors showed how to detect and monitor tumor-derived mutations in circulating DNA in NSCLC patients using the UltraSEEK Lung Panel. They concluded that the UltraSEEK Lung Panel is an alternative tool to detect mutations at diagnosis and disease progression, as well as to monitor treatment response.
I only have 2 minor issues to be addressed prior to acceptance of the manuscript:
1) The quality of figure 2 should be improved
2) Statistical analyses performed should be showed in all figure legends. Authors should also show the number of stars (significance) in the graph instead of p<0.05 (Figure 3)
Author Response
In this interesting study, authors showed how to detect and monitor tumor-derived mutations in circulating DNA in NSCLC patients using the UltraSEEK Lung Panel. They concluded that the UltraSEEK Lung Panel is an alternative tool to detect mutations at diagnosis and disease progression, as well as to monitor treatment response.
I only have 2 minor issues to be addressed prior to acceptance of the manuscript:
1) The quality of figure 2 should be improved
Answer to reviewer: We thank the reviewer for this comment. We have improved the quality of all figures significantly.
2) Statistical analyses performed should be showed in all figure legends. Authors should also show the number of stars (significance) in the graph instead of p<0.05 (Figure 3)
Answer to reviewer: The statistical analyses performed are now mentioned in the figure legends. The strength of significance is indicated in Figure 3 as Figure 3C displays P<0.05 and Figure 3C shows P<0.01, equal to * and **, respectively.